environmental chemistry/green chemistry/ environmental engineering

flotation reagents, thiol collector, UV$_{185+254\,nm}$ photolysis, vacuum-UV, mineralization, sulfur byproducts

**Author for correspondence:**
Pingfeng Fu
e-mail: pffu@ces.ustb.edu.cn

This article has been edited by the Royal Society of Chemistry, including the commissioning, peer review process and editorial aspects up to the point of acceptance.

# UV$_{185+254\,nm}$ photolysis of typical thiol collectors: decomposition efficiency, mineralization and formation of sulfur byproducts

Pingfeng Fu[1,2], Gen Li[1], Xiaoting Wu[1], Xiaofeng Lin[1] and Bolan Lei[1]

[1]School of Civil and Resources Engineering, University of Science and Technology Beijing, Beijing 100083, People's Republic of China
[2]Key Laboratory of High-efficient Mining and Safety of Metal Mines, Ministry of Education, Beijing 100083, People's Republic of China

(iD) PF, 0000-0002-6956-1004

The decomposition of toxic flotation reagents upon UV$_{185+254\,nm}$ irradiation was attractive due to operational simplicity and no dosage of oxidants. In this work, the degradation of typical thiol collectors (potassium ethyl xanthate (PEX), sodium diethyl dithiocarbamate (SDD), *O*-isopropyl-*N*-ethyl thionocarbamate (IET) and dianilino dithiophoshoric acid (DDA)) was investigated by UV$_{185+254\,nm}$ photolysis. The degradation efficiencies and mineralization extents of collectors were assessed. The formation of $CS_2$ and $H_2S$ byproducts was studied, and the mechanisms of collector degradation were proposed under UV$_{185+254\,nm}$ irradiation. The PEX, SDD and IET were decomposed with nearly 100% removal upon 75 min of UV$_{185+254\,nm}$ irradiation. The decomposition rate constants decreased in the order SDD > PEX > IET $\gg$ DDA, and the DDA was the refractory collector. After 120 min of UV$_{185+254\,nm}$ irradiation, $15-45\%$ of carbon and $25-75\%$ of sulfur of collectors were completely mineralized, and the mineralization extent decreased in the order PEX > SDD > IET > DDA. The percentage of gaseous sulfur ($CS_2$ and $H_2S$) ranged from 0.48 to 4.85% for four collectors, showing the fraction of emitted sulfur byproducts was small. The aqueous $CS_2$ concentration increased in the first $10-20$ min, and was decreased to a low level of $0.05-0.1\,mg\,l^{-1}$ at 120 min. Two mechanisms, i.e. direct UV$_{254\,nm}$ photolysis and indirect oxidation with free radicals, were responsible for collector decomposition in the UV$_{185+254\,nm}$ photolysis.

# 1. Introduction

Froth flotation has become the most widely applied process for separating valuable minerals from ores in mines around the world. Thiol collectors, such as xanthates, dithiophosphates and dithiocarbamates, are important flotation reagents to render sulfide minerals hydrophobic and facilitate bubble attachments [1,2]. To achieve high recovery of non-ferrous metals, the dosage of collectors is frequently ranged from 30 to 300 g ton$^{-1}$ (ore). Therefore, the consumption of thiol collectors becomes very large due to the extremely high amount of treated ores. Even in the 1980s, the global consumption of xanthates had reached above 52 000 tons per year [2]. Nevertheless, nearly 50% of collectors dosed in flotation circuits would be discharged in wastewaters after the mineral flotation [3]. Some collectors and their byproducts are found to be toxic to soil microbes, biota, animals and human beings [4–7]. Previously, the hazards of xanthate to frog embryos have been reviewed [7]. Accordingly, the discharge of collectors from mineral flotation may cause serious environmental pollution. Therefore, it has been of great concern to remove potentially toxic collectors from flotation wastewaters for the sustainable development of mining industry.

Recently, some processes have been developed to remove organic reagents from flotation wastewaters, including adsorption [8], chemical oxidation [9,10], ozonation [11,12] and biodegradation [13,14]. Chemical oxidation and ozonation have exhibited high efficiency in decomposing flotation reagents. But these processes need the dosage of chemical oxidants, such as sodium hypochlorite [9], persulfate [10], hydrogen peroxide [15] and ozone [11]. The usage of highly reactive oxidants in mines makes them less attractive and sometimes expensive. Since most of the mines are located far from industrial zones, there are potential risks associated with long distance transportation and storage of dangerous chemicals. The biodegradation is widely accepted as a low cost process in wastewater treatment. However, previous reports indicate that some flotation reagents are toxic to microbes, considerably reducing microbial activities in the biodegradation of flotation reagents [13,16].

Advanced oxidation processes (AOPs) involving ozone [10], Fenton's reagent [17], hydrogen peroxide [15], persulfate [18,19] and photocatalyst [20–22] have been investigated to effectively degrade flotation collectors, pharmaceuticals and textile dyes. However, most of these studies only concerned the removal efficiencies of xanthates [11,12,15]. Up to now, research on the removal of other thiol collectors such as dithiophosphates and dithiocarbamates has been scarce. Because thiol collectors have different molecular structures [1,23], their decomposition behaviours by the same AOP method may differ. Additionally, the functional groups of all thiol collectors have sulfur atoms [1,23]. So, it is possible to generate toxic sulfur byproducts such as $CS_2$ and $H_2S$, while organic sulfur of collectors is mineralized to sulfate. For example, the concentrations of aqueous $CS_2$ were determined in the ozonation of xanthates [24]. However, the generation of sulfur byproducts receives little concern in the decomposition of thiol collectors by the AOPs [10,12].

The UV-based AOPs, such as $UV_{254\,nm}/O_3$ and $UV_{254\,nm}/TiO_2$ photocatalysis, can effectively decompose flotation collectors and subsequently mineralize various byproducts [11,20]. But these AOPs involve the dosage of oxidants or powder catalysts, making them expensive and complex. The $UV_{185+254\,nm}$ photolysis, irradiated by a low-pressure Hg lamp emitting both vacuum-UV (VUV) at 185 nm and UVC at 254 nm, is a simple and promising AOP. The 185 nm VUV irradiation of water can directly generate hydroxyl radicals (OH•), hydrogen radicals (H•), solvated free electrons ($e_{aq}^-$), superoxide radicals (HO$_2$•, $O_2^{\bullet-}$) and $H_2O_2$ as shown in equations (1.1)−(1.6) [25]. The formation of OH• radicals and superoxide anion ($O_2^{\bullet-}$) in the VUV irradiation of water have been proven by the electron paramagnetic resonance spectroscopy [26]. The $UV_{185+254\,nm}$ photolysis has shown high oxidation capacity for pollutants, such as odour compounds [27], pharmaceuticals [28] and pesticides [29]. Moreover, the operation of $UV_{185+254\,nm}$ photolysis is much simpler than the $UV_{254\,nm}/O_3$ and $UV_{254\,nm}/TiO_2$ photocatalysis because no oxidant or catalyst is required. In terms of the practical treatment of flotation wastewaters, the transportation and storage of dangerous oxidants in mines can also be avoided if the $UV_{185+254\,nm}$ photolysis is applied. Thus, it is quite necessary to assess the removal performances of flotation reagents by the $UV_{185+254\,nm}$ photolysis. However, as far as we know, there is no report on the $UV_{185+254\,nm}$ photolysis of organic flotation reagents

$$H_2O + h\nu_{185nm} \rightarrow OH\bullet + H^+ + e_{aq}^-, \tag{1.1}$$

$$H_2O + h\nu_{185nm} \rightarrow OH\bullet + H\bullet, \tag{1.2}$$

$$2OH\bullet \rightarrow H_2O_2, \tag{1.3}$$

$$H\bullet + O_2 \rightarrow HO_2\bullet, \tag{1.4}$$

**Table 1.** Collector name, molecular structure, molecular formula and abbreviation.

| collector name | molecular structure | molecular formula | abbreviation |
|---|---|---|---|
| potassium ethyl xanthate | $CH_3-CH_2-O-\overset{\overset{S}{\|\|}}{C}-S-K$ | $C_2H_4OCSSK$ | PEX |
| sodium diethyl dithiocarbamate | $\begin{smallmatrix}CH_3-CH_2\\ \quad\quad\quad N-\overset{\overset{S}{\|\|}}{C}-S-Na\\ CH_3-CH_2\end{smallmatrix}$ | $(C_2H_5)_2NCSSNa$ | SDD |
| O-isopropyl-N-ethyl thionocarbamate | $H-\overset{\overset{CH_3}{\|}}{\underset{\underset{CH_3}{\|}}{C}}-O-\overset{\overset{S}{\|\|}}{C}-\overset{\overset{H}{\|}}{N}-CH_2-CH_3$ | $(CH_3)_2CHOCSNHC_2H_5$ | IET |
| dianilino dithiophoshoric acid | (structure: two phenyl-NH groups bonded to P, with P=S and P−S−H) | $(C_6H_5NH)_2PSSH$ | DDA |

$$HO_2\bullet + H_2O \Leftrightarrow H_3O^+ + O_2^{\bullet-} \quad pK_a = 4.8 \tag{1.5}$$

and

$$2HO_2\bullet \rightarrow O_2 + H_2O_2. \tag{1.6}$$

In this work, the $UV_{185+254\,nm}$ photolysis of thiol collectors is investigated by using a 40 W low-pressure Hg lamp, which emits about 10% radiation at 185 nm and 90% radiation at 254 nm. Four thiol collectors, potassium ethyl xanthate (PEX), sodium diethyl dithiocarbamate (SDD), O-isopropyl-N-ethyl thionocarbamate (IET) and dianilino dithiophoshoric acid (DDA), are selected as typical sulfide mineral collectors. The objectives of this work are (i) to assess the feasibility of decomposing thiol collectors by the $UV_{185+254\,nm}$ photolysis, (ii) to determine the generation of $CS_2$ and $H_2S$ byproducts, as well as (iii) to propose the collector decomposition mechanisms under $UV_{185+254\,nm}$ irradiation. The mineralization of collectors is examined by measuring total organic carbon (TOC) and $SO_4^{2-}$ concentrations. The amounts of gaseous $CS_2$ and $H_2S$, as well as the concentration of aqueous $CS_2$, were measured to study the generation of sulfur byproducts.

# 2. Material and methods

## 2.1. UV lamps and chemicals

Two types of low-pressure Hg lamps with electrical power of 40 W were purchased from Bright Star Light & Electricity Co., Ltd, Guangdong, China. The $UV_{185+254\,nm}$ lamp transmitted both 185 nm VUV and 254 nm UV light through high-purity quartz glass, but the $UV_{254\,nm}$ lamp only transmitted 254 nm UV light. The $UV_{254\,nm}$ irradiation intensity of $UV_{185+254\,nm}$ lamp was about 90% of $UV_{254\,nm}$ lamp. The lamps had a length of 1199 mm and a diameter of 19 mm.

The PEX and SDD with analytical grade were purchased from Aladdin Chemical Reagent Co., Ltd, Shanghai, China. The industrial grade IET and DDA were purchased from Tieling Flotation Reagents Co., Ltd, Liaoning, China. Their molecular structures, molecular formulas and abbreviations are summarized in table 1. Other chemicals such as cupric acetate, lead acetate, diethylamine, triethanolamine, silver sulfate and mercury sulfate were of analytical grade and purchased from Sinopharm Chemical Reagent Co., Ltd, Beijing, China. Deionized water was used in the degradation experiments.

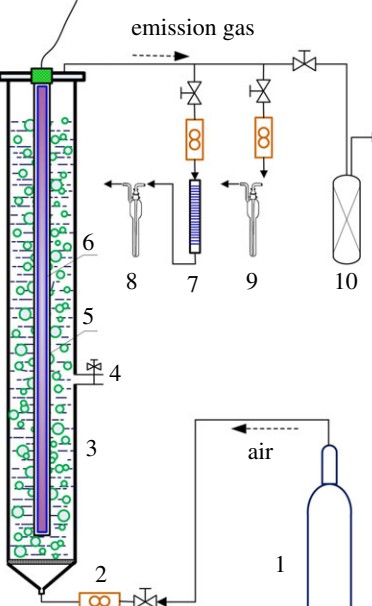

**Figure 1.** Schematic diagram of experimental set-up. (1, air bottle; 2, flow meter; 3, purged reactor; 4, sampling valve; 5, quartz tube; 6, $UV_{185+254\text{ nm}}$ or $UV_{254\text{ nm}}$ lamp; 7, absorbent cotton with lead acetate; 8, $CS_2$ absorption liquid; 9, $H_2S$ absorption liquid; 10, activated carbon column.)

## 2.2. Experimental set-up and degradation procedures

All degradation experiments were conducted with a batch mode in a jacket glass reactor connected to a thermostatic bath. The schematic diagram of the experimental set-up is shown in figure 1. The cylindrical reactor, with 1240 mm height and 53 mm internal diameter, was installed at its axis with a quartz tube with a height of 1220 mm and outer diameter of 21 mm. The $UV_{185+254\text{ nm}}$ or $UV_{254\text{ nm}}$ lamp was inserted into the quartz tube. The optical distance from tube surface to the internal surface of the reactor was 16 mm. The air was continuously purged into the reactor through a porous glass plate with a flow rate of $1.67\text{ l min}^{-1}$. The injected air stream not only provided dissolved oxygen for degradation reactions, but also stirred the collector solutions. The degradation experiments were conducted at $25 \pm 2°\text{C}$.

Prior to photolysis experiments, the collector (0.2 g) was dissolved in 2 l deionized water to prepare a PEX (SDD, IET or DDA) solution of $100\text{ mg l}^{-1}$ concentration. The initial pH was adjusted to $7.0-12.0$ using $0.05\text{ mol l}^{-1}$ NaOH or HCl solution. When the collector solution was introduced into the reactor purged with an air stream, the $UV_{185+254\text{ nm}}$ or $UV_{254\text{ nm}}$ lamp was turned on to carry out photolysis experiments. The irradiation time was controlled to be 120 min. Because $CS_2$, $H_2S$ and other sulfur byproducts were emitted into gas phase, the emission gas was introduced into an activated carbon column to remove toxic byproducts. The aqueous samples were taken at designated intervals to determine the concentrations of collector, chemical oxygen demand (COD), TOC, $CS_2$ and $SO_4^{2-}$ ions.

## 2.3. Analysis and calculation

### 2.3.1. Determination of collector concentration, COD and TOC

The concentration of thiol collectors (PEX, SDD, IET and DDA) was determined by a UV–vis spectroscopic method [10,11,13,30]. The absorbance of collector solution was recorded by a UV–vis spectrophotometer (UV-5500PC, Shanghai Metash Instruments Co. Ltd, China). The COD was determined by the standard dichromate method (HJ/T 399-2007). The TOC of water samples was measured by using a Shimadzu TOC-V organic carbon analyzer. The carbon mineralization extent ($\gamma_c$) of the collector (PEX, SDD, IET and DDA) was calculated as the below equation

$$\gamma_c = \frac{\text{TOC}_0 - \text{TOC}_t}{\text{TOC}_0} \times 100\%, \qquad (2.1)$$

where $\text{TOC}_0$ and $\text{TOC}_t$ ($\text{mg l}^{-1}$) were the TOC concentration at initial and time $t$, respectively.

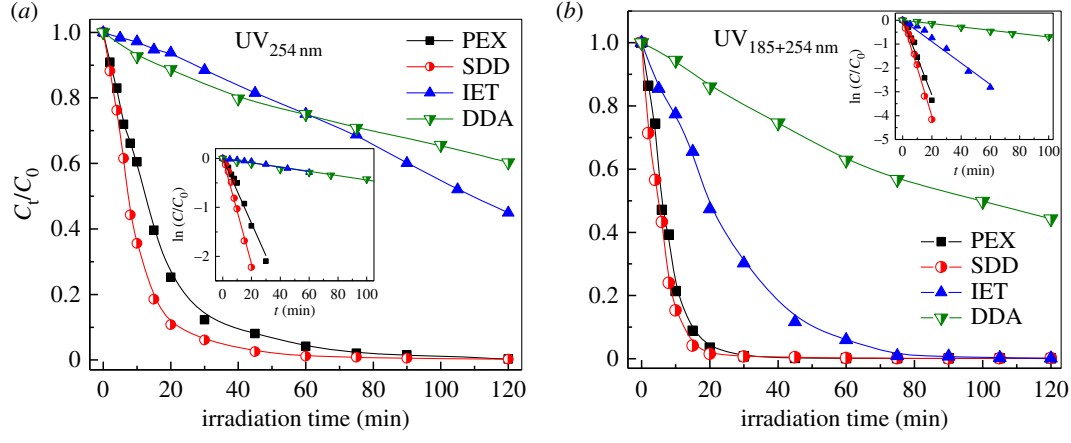

**Figure 2.** The variations of relative concentration ($C_t/C_0$) with irradiation time upon $UV_{254 nm}$ (*a*) and $UV_{185+254 nm}$ (*b*) irradiation. The insets are the pseudo-first-order kinetic fitting of $\ln(C_t/C_0)$ versus time $t$. Experimental conditions: collector concentration of 100 mg l$^{-1}$ and initial pH of 10.0.

### 2.3.2. Determination of concentrations of aqueous $CS_2$ and $SO_4^{2-}$ ions

The aqueous $CS_2$ concentration was measured by a diethylamine cupric acetate spectrophotometric method (GB/T 15504-1995). Fifty millilitres of water sample were continuously purged by a 100 ml min$^{-1}$ $N_2$ stream for 1 h to volatilize $CS_2$, which was absorbed by a diethylamine and cupric acetate mixed liquid. Then the absorbance of absorption liquid was measured at 430 nm using dehydrated alcohol as a reference solution. The concentration of sulfate was determined by a barium chromate spectrophotometry method (HJ/T 342-2007). Because $SO_4^{2-}$ ions, with the highest valence of sulfur, were the final oxidization product of organic sulfur in collectors, the sulfur mineralization extent ($\gamma_s$) of the collector was defined as the below equation [31]

$$\gamma_s = \frac{M}{n \times 96} \times \frac{C_{SO_4^{2-},t}}{C_0} \times 100\% \tag{2.2}$$

where $M$ and $n$ were the molecular weight and number of sulfur atom in the collector (PEX, SDD, IET or DDA), respectively, $C_{SO_4^{2-},t}$ (mg l$^{-1}$) was the concentration of $SO_4^{2-}$ ions at time $t$, and $C_0$ (mg l$^{-1}$) was the initial concentration of the collector.

### 2.3.3. Measurement of the amount of gaseous $CS_2$ and $H_2S$

The amount of gaseous $CS_2$ emitted from the reactor for 120 min was measured by a diethylamine spectrophotometric method (GB/T 14680-93). By adding 5.0 mg cupric acetate, 2.5 ml diethylamine and 2.5 ml triethanolamine into a 500 ml volumetric flask, the $CS_2$ absorption liquid was prepared by diluting the mixture with dehydrated alcohol to 500 ml. As shown in figure 1, before the emission gas was introduced into $CS_2$ absorption liquid, it was first passed through a glass tube filled by absorbent cotton coated with lead acetate to remove $H_2S$. The amount of gaseous $H_2S$ for 120 min was measured by a methylene blue spectrophotometric method (GB/T 11742-89). By adding 4.3 g cadmium sulfate, 0.30 g NaOH and 10.0 g polyvinyl alcohol ammonium phosphate into a 1000 ml volumetric flask, the $H_2S$ absorption liquid was prepared by diluting the mixture with deionized water to 1000 ml.

# 3. Results and discussion

## 3.1. Effect of UV wavelength

The photolysis of thiol collectors (PEX, SDD, IET and DDA) upon $UV_{254 nm}$ or $UV_{185+254 nm}$ irradiation is shown in figure 2. The removal efficiency of collectors and decomposition rate constants are summarized in table 2. In this work, overall kinetics of the collector degradation can be described by a simple pseudo-first-order rate law [32]. As shown in figure 2*a* and table 2, the removal efficiency of PEX and SDD at 75 min reached 97.93% and 99.16%, respectively, which was much higher than that (about 30%) of

**Table 2.** The removal efficiencies of collectors, decomposition rate constants ($k_{collector}$), half-lives ($t_{1/2}$) and correlation coefficients ($R^2$) in the degradation of thiol collectors under $UV_{185+254\ nm}$ and $UV_{254\ nm}$ irradiation, respectively.

| collectors | UV irradiation | removal efficiency (%) | | $k_{collector}$ (min$^{-1}$) | $t_{1/2}$ (min) | $R^2$ |
| --- | --- | --- | --- | --- | --- | --- |
| | | 20 min | 75 min | | | |
| PEX | $UV_{185+254\ nm}$ | 96.51 | 99.86 | 0.1565 | 4.43 | 0.982 |
| | $UV_{254\ nm}$ | 74.77 | 97.93 | 0.06629 | 10.46 | 0.989 |
| SDD | $UV_{185+254\ nm}$ | 98.45 | 99.88 | 0.2005 | 3.02 | 0.990 |
| | $UV_{254\ nm}$ | 89.21 | 99.16 | 0.1076 | 6.44 | 0.992 |
| IET | $UV_{185+254\ nm}$ | 52.62 | 99.08 | 0.04485 | 15.45 | 0.986 |
| | $UV_{254\ nm}$ | 6.25 | 31.20 | 0.00447 | 155.07 | 0.988 |
| DDA | $UV_{185+254\ nm}$ | 13.98 | 43.26 | 0.00701 | 98.88 | 0.998 |
| | $UV_{254\ nm}$ | 11.31 | 29.17 | 0.00438 | 158.25 | 0.991 |

IET and DDA upon $UV_{254\ nm}$ irradiation. In UV-irradiated solutions, the decomposition of pollutants is frequently initiated by excited molecules with absorbing UV irradiation [33]. Among four collectors, the PEX and SDD may have higher mole absorptivity of $UV_{254\ nm}$ irradiation than IET and DDA, resulting in higher removal efficiencies. In most mines, residual flotation reagents are usually removed in tailing ponds by a natural degradation process involving sunlight and oxygen. As given in table 2, the half-life of IET and DDA was almost one order of magnitude larger than that of PEX and SDD. Therefore, it can reasonably be inferred that after the natural degradation, residual IET and DDA in tailing ponds would have higher concentrations than the PEX and SDD.

As exhibited in figure 2b and table 2, the removal efficiency of PEX and SDD had reached above 96% even at 20 min under $UV_{185+254\ nm}$ irradiation. At 75 min, nearly 100% removal of PEX, SDD and IET was achieved, revealing that most thiol collectors could be effectively degraded by the $UV_{185+254\ nm}$ photolysis without adding any oxidant. The decomposition rate constant ($k_{collector}$) of collectors upon $UV_{185+254\ nm}$ irradiation decreased in the order $k_{SDD}$ (0.2005 min$^{-1}$) $> k_{PEX}$ (0.1565 min$^{-1}$) $> k_{IET}$ (0.04485 min$^{-1}$) $\gg k_{DDA}$ (0.00701 min$^{-1}$). As listed in table 2, the $k_{collector}$ values for four collectors under $UV_{185+254\ nm}$ irradiation were 1.60−10.03 times higher than those achieved upon $UV_{254\ nm}$ irradiation, showing the enhanced degradation of thiol collectors by the $UV_{185+254\ nm}$ photolysis. However, the DDA, with very low $k_{collector}$ value, can be considered as a refractory collector for the $UV_{185+254\ nm}$ photolysis.

## 3.2. Effect of initial pH

The flotation of sulfide minerals is usually conducted in alkaline pulps. Thus, residual collectors are present in the flotation wastewaters at pH range from neutral to alkaline [34]. In this work, the effect of initial pH on the $UV_{185+254\ nm}$ photolysis of PEX and DDA was investigated. As shown in figure 3, both the $k_{PEX}$ and $k_{DDA}$ values gradually decreased as initial pH rose. The decomposition rate constants obtained at pH 7.0 were 2.4 times for PEX and 4.9 folds for DDA higher than those achieved at pH 12.0, respectively. It suggests that the neutral pH can facilitate the decomposition of flotation collectors upon $UV_{185+254\ nm}$ irradiation. In the previous studies, the higher decomposition efficiencies were also achieved at lower pH during the $UV_{185+254\ nm}$ photolysis of 1,4-dioxane [30] and 4-*tert*-octylphenol [35].

Upon 185 nm VUV irradiation, the homolysis and photochemical ionization of $H_2O$ occur with the generation of OH• radicals (equations (1.1) and (1.2)), and OH• radicals (or $HO_2$•) can recombine to form $H_2O_2$ (equations (1.3) and (1.6)) [25]. Under $UV_{185+254\ nm}$ irradiation, $H_2O_2$ can be decomposed by absorbing $UV_{254\ nm}$ light to form OH• radicals (equation (3.1)). Accordingly, OH• radicals and $H_2O_2$ are in equilibrium under UV irradiation offered by the $UV_{185+254\ nm}$ lamp. On the other hand, the concentration of $H_2O_2$ is also dependent on the pH value of the reaction system. As given in equation (3.2), $H_2O_2$ itself is in equilibrium with OH$^-$ anions [36]. Additionally, $HO_2$• radicals are also in equilibrium with protons ((equation (1.5)). According to equations (1.5), (1.6) and (3.2), it can be seen that high H$^+$ concentration (i.e. low pH) promotes the equilibrium to form $H_2O_2$. In turn,

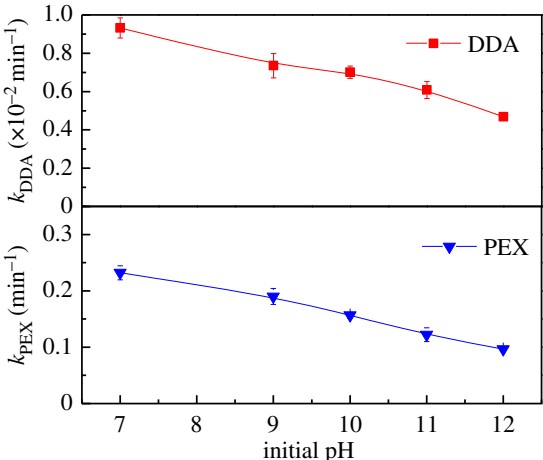

**Figure 3.** The variations of the $k_{PEX}$ and $k_{DDA}$ with initial pH under $UV_{185+254\ nm}$ irradiation. Experimental conditions: collector concentration of 100 mg l$^{-1}$.

increased concentration of $H_2O_2$ results in the formation of more OH• radicals (equation (3.1)). This equilibrium consideration may explain why the $k_{PEX}$ and $k_{DDA}$ values were higher at the neutral pH under $UV_{185+254\ nm}$ irradiation

$$H_2O_2 + h\nu_{254\,nm} \rightarrow OH\bullet + OH\bullet \tag{3.1}$$

and

$$H_2O_2 + OH^- \Leftrightarrow H_2O + HO_2^- \qquad pK_a = 11.62. \tag{3.2}$$

In addition, the equilibrium of carbonate may also influence the pH dependence of the collector decomposition. The mineralization of contaminants by UV-based AOPs results in the formation of $CO_2$, subsequently increasing the aqueous concentration of $CO_3^{2-}$ anions. According to the equilibrium of $CO_3^-$ in water, $HCO_3^-$ and $CO_3^{2-}$ anions dominate the equilibrium approximately at neutral to weak basic pH ($6 < pH < 10$) and at strong basic pH ($pH > 10$), respectively. These carbonate species can scavenge OH• radicals. However, $CO_3^{2-}$ anions have a larger scavenging capacity than $HCO_3^-$ [37]. Thus, at higher initial pH, more OH• radicals would be scavenged by $CO_3^{2-}$ anions, resulting in lower decomposition rate constants of PEX and DDA.

## 3.3. The mineralization of thiol collectors

To investigate the mineralization of thiol collectors under $UV_{185+254\ nm}$ irradiation, the concentrations of COD, TOC and $SO_4^{2-}$ ions were measured with the results shown in figure 4. For each collector, the relative concentrations of COD and TOC declined, and the $SO_4^{2-}$ concentration increased as the collector was decomposed. It suggested that the byproducts had been further degraded with the formation of $CO_2$ and $SO_4^{2-}$ ions under $UV_{185+254\ nm}$ irradiation. As indicated in figure 4, the TOC was reduced just a bit while the evident decrease in COD was observed for each collector. Thus, the removal efficiency of COD was higher than the mineralization extent of carbon as given in table 3. By comparing the decomposition rate constants for collector and COD removal, as summarized in tables 2 and 3, the $k_{collecor}$ for each collector was $3-21$ folds higher than the $k_{COD}$ value. It clearly indicated that only a small fraction of organic carbon in collectors was mineralized to $CO_2$ upon $UV_{185+254\ nm}$ irradiation although nearly 100% removal of PEX, SDD and IET was achieved. For four collectors, the mineralization extent of carbon ($\gamma_c$) at 120 min increased in the order DDA < IET < SDD < PEX. In particular, the $\gamma_c$ for the PEX was much higher than that for SDD, IET and DDA, indicating that the byproducts derived from PEX seemed to be readily decomposed by OH• radicals.

As shown in figure 4, the concentration of $SO_4^{2-}$ ions increased up to 90.36 mg l$^{-1}$ for PEX, 64.79 mg l$^{-1}$ for SDD, 26.90 mg l$^{-1}$ for IET and 11.48 mg l$^{-1}$ for DDA, respectively, under $UV_{185+254\ nm}$ irradiation for 120 min. The $SO_4^{2-}$ ions were the final sulfur product in the oxidation of organic sulfur in thiol collectors [11,12,24]. The occurrence of $SO_4^{2-}$ ions with increased concentrations well demonstrated that organic sulfur of collectors could be completely mineralized [24]. As summarized in table 3, the mineralization extent of sulfur ($\gamma_s$) at 120 min had reached 74.83% for PEX, 57.69% for SDD, 41.19% for IET and

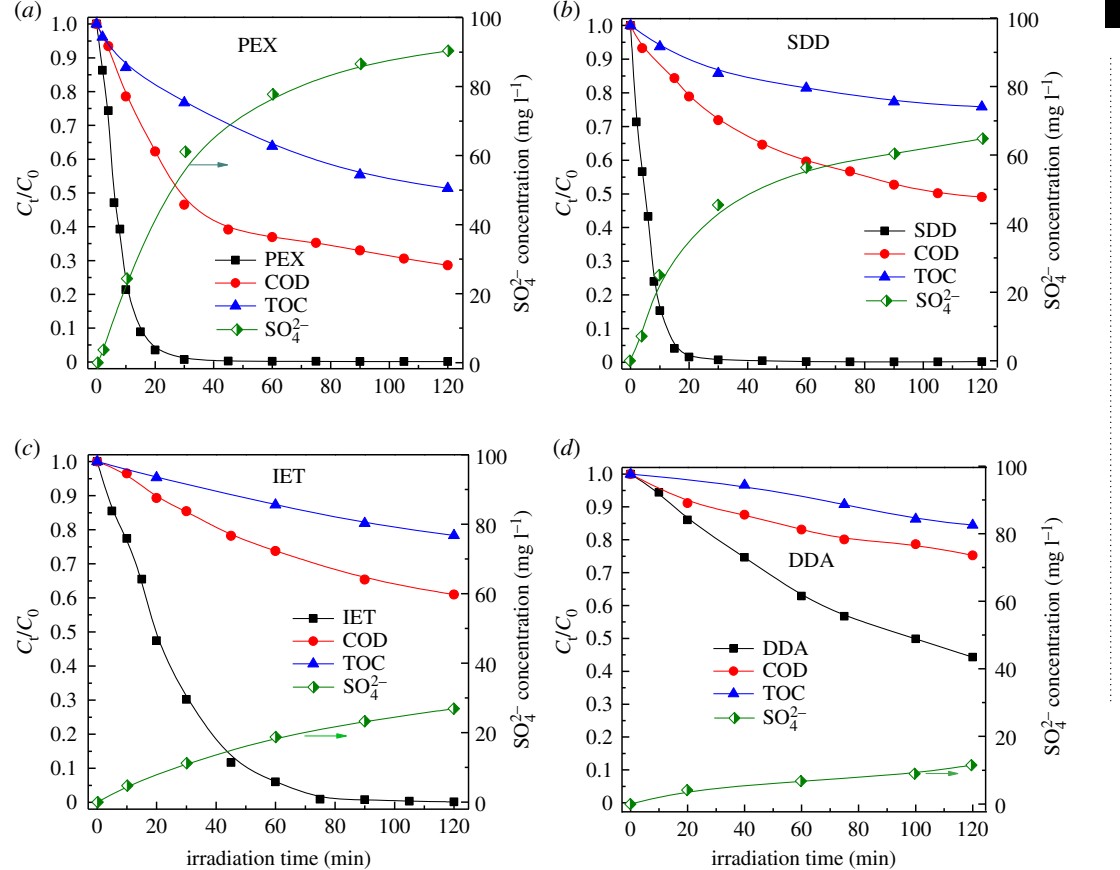

**Figure 4.** The variations of $SO_4^{2-}$ concentration and relative concentrations of collector (PEX (a), SDD (b), IET (c) and DDA (d)), COD and TOC with $UV_{185+254\ nm}$ irradiation time. Experimental conditions: collector concentration of 100 mg $l^{-1}$ and initial pH of 10.0.

**Table 3.** The removal efficiencies of COD, decomposition rate constants for COD removal ($k_{COD}$), correlation coefficients ($R^2$) and the mineralization extents of carbon and sulfur in the $UV_{185+254\ nm}$ photolysis of thiol collectors.

| collector | removal of COD | | | mineralization extent (%) | |
|---|---|---|---|---|---|
| | removal efficiency of COD (%) | $k_{COD}$ (min$^{-1}$) | $R^2$ | carbon | sulfur |
| PEX | 71.40 | 0.02245 | 0.9896 | 48.62 | 74.83 |
| SDD | 50.92 | 0.00953 | 0.9844 | 24.17 | 57.69 |
| IET | 39.04 | 0.00455 | 0.9878 | 21.68 | 41.19 |
| DDA | 24.77 | 0.00259 | 0.9783 | 15.56 | 16.75 |

16.75% for DDA, respectively. Thiol collectors except for DDA had much higher $\gamma_s$ values than the $\gamma_c$, suggesting that the conversion of organic sulfur to $SO_4^{2-}$ was more efficient than the mineralization of organic carbon to $CO_2$. By considering $\gamma_c$ and $\gamma_s$ together, the mineralization extent of thiol collectors decreased in the order PEX > SDD > IET > DDA upon $UV_{185+254\ nm}$ irradiation. In particular, the PEX and SDD had much larger extent of mineralization when compared with IET and DDA at the same degradation conditions.

## 3.4. Generation of CS$_2$ and H$_2$S byproducts

Sulfur byproducts, such as carbon disulfide ($CS_2$), carbonyl sulfide (COS), hydrogen sulfide ($H_2S$), sulfite ($SO_3^{2-}$) and thiosulfate ($S_2O_3^{2-}$), were detected in the oxidization of xanthates by the AOPs [15,24,38–40]. Among these sulfur byproducts, the $CS_2$, COS and $H_2S$ are toxic and highly volatile contaminants [41].

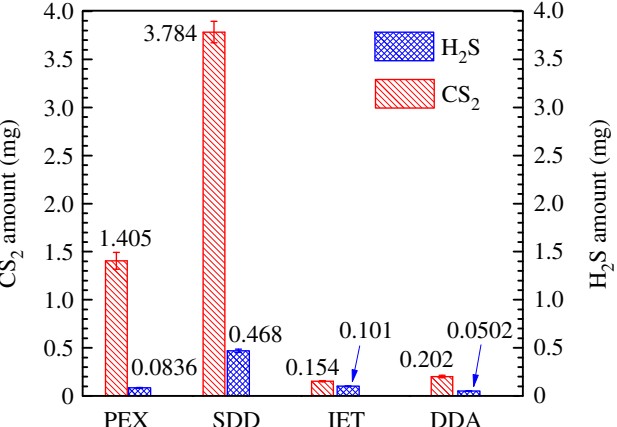

**Figure 5.** The amounts of emitted $CS_2$ and $H_2S$ in gas phase by the $UV_{185+254 \text{ nm}}$ photolysis of thiol collectors for 120 min. Experimental conditions: collector concentration of 100 mg l$^{-1}$ and initial pH of 10.0.

For example, $CS_2$ is considered as a hazard air pollutant under the Title III of the 1990 Clean Air Act Amendment (CAAA) of the USA [42]. Accordingly, while toxic sulfur byproducts are emitted from air purged flotation wastewaters into the atmosphere or indoor environment, it may pose a significant hazard to safety, health and the environment (SHE). However, to the best of our knowledge, no work is available in the literature on the quantitative determination of emitted $CS_2$ and $H_2S$ in gas phase when thiol collectors are decomposed by the AOPs. For the aqueous $CS_2$ byproduct, Fu *et al.* [11,24] had measured the $CS_2$ concentration while degrading xanthates by the $O_3$ and $UV/O_3$ processes. However, the evolutions of aqueous $CS_2$ byproduct has not yet been investigated during the degradation of thiol collectors, except for xanthates by the AOPs.

In this work, the amounts of gaseous $CS_2$ and $H_2S$ emitted from air purged solutions were measured using the $CS_2$ and $H_2S$ absorption liquids, respectively. As illustrated in figure 5, upon $UV_{185+254 \text{ nm}}$ irradiation of 100 mg l$^{-1}$ collector solutions for 120 min, the amount of gaseous $CS_2$ reached 1.405 mg for PEX, 3.784 mg for SDD, 0.154 mg for IET and 0.202 mg for DDA, respectively. The results indicated that the SDD and PEX released much larger amount of gaseous $CS_2$ than IET and DDA. For the emission of $H_2S$ byproduct, gaseous $H_2S$ reached 0.0836 mg for PEX, 0.468 mg for SDD, 0.101 mg for IET and 0.0502 mg for DDA, respectively. Among four collectors, the SDD had released the largest amount of $H_2S$ into gas phase. For each collector, the amount of gaseous $CS_2$ released was larger than that of $H_2S$, and the trends became more evident for PEX and SDD. By considering $CS_2$ and $H_2S$ together, the amount of gaseous sulfur byproducts released upon $UV_{185+254 \text{ nm}}$ irradiation increased in the order IET ≈ DDA < PEX < SDD. According to these results, it could be seen that the release of gaseous sulfur byproducts was quite diverse due to their different molecular structures and sulfur contents of thiol collectors.

As mentioned above, the emitted $CS_2$ and $H_2S$ are toxic gases. Thus, it is essential to indicate the percentage of gaseous sulfur to total sulfur in collectors. By assuming that no volatile sulfur byproducts except for $CS_2$ and $H_2S$ are present in emission gas, the percentage of gaseous sulfur ($\beta_{s,g}$) can be defined according to the below equation

$$\beta_{s,g} = \frac{m_1 \times (64/76) + m_2 \times (32/34)}{((n \times 32)/M) \times C_0 \times V} \times 100\%, \tag{3.3}$$

where $m_1$ and $m_2$ (mg) were the amount of emitted $CS_2$ and $H_2S$ into gas phase for 120 min, $n$ was the number of S atom in collectors and $V$ (l) was the volume of collector solutions. Thus, the $\beta_{s,g}$ values for four collectors are summarized in table 4. Except for the SDD, the $\beta_{s,g}$ for PEX, IET and DDA was very low, showing that only a small fraction of sulfur in thiol collector was released into emission gas in the $UV_{185+254 \text{ nm}}$ photolysis. However, a previous study had estimated that 20.6% of total sulfur in *n*-butyl xanthate was released into gas phase after the oxidation by $O_3$ [12]. These data were achieved by subtracting total sulfur in *n*-butyl xanthate with sulfur in $SO_4^{2-}$ ions. Since other sulfur byproducts such as sulfite, thiosulfate as well as organic sulfur compounds would be also presented in aqueous solutions, 20.6% of sulfur released in gas phase might be overestimated by Yan *et al.* [12].

In this work, aqueous $CS_2$ concentrations were also measured. As presented in figure 6, the $CS_2$ concentration for each collector rapidly increased to its maximum value in the first 10−20 min, and

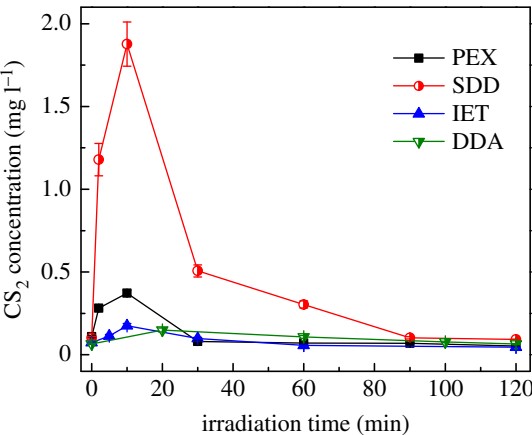

**Figure 6.** The variations of aqueous $CS_2$ concentration with $UV_{185+254\ nm}$ irradiation time. Experimental conditions: collector concentration of 100 mg l$^{-1}$ and initial pH of 10.0.

**Table 4.** The percentage of gaseous sulfur ($\beta_{s,g}$) in the $UV_{185+254\ nm}$ photolysis of thiol collectors for 120 min.

| collector | PEX | SDD | IET | DDA |
|---|---|---|---|---|
| $\beta_{s,g}$ | 1.57 | 4.85 | 0.53 | 0.48 |

then gradually decreased to a low level under $UV_{185+254\ nm}$ irradiation. The measured maximum concentration of $CS_2$ was 0.37 mg l$^{-1}$ for PEX, 1.88 mg l$^{-1}$ for SDD, 0.18 mg l$^{-1}$ for IET and 0.15 mg l$^{-1}$ for DDA, respectively. It can be clearly seen that the decomposition of SDD can generate much more $CS_2$ in comparison to other collectors. Additionally, the order of maximum $CS_2$ concentration for four collectors was well consistent with that for the amount of gaseous $CS_2$ as shown in figure 5. As shown in figure 6, residual $CS_2$ concentrations for all collectors were decreased to a low level of 0.05−0.1 mg l$^{-1}$ after 120 min of $UV_{185+254\ nm}$ irradiation. By considering the remarkable increase in $SO_4^{2-}$ concentrations shown in figure 4, it can be reasonably inferred that most of $CS_2$ byproduct was converted to $SO_4^{2-}$ ions by OH• radicals with the pathways as elucidated in our previous work [24].

As discussed above, the decomposition of SDD and PEX had generated a larger amount of $CS_2$ in comparison to IET and DDA. As summarized in table 1, both the sulfur content and molecular structure of thiol collectors are quite different from each other. Accordingly, the different sulfur contents and decomposition mechanisms caused by different molecular structures would be associated with the formation of sulfur byproducts under $UV_{185+254\ nm}$ irradiation. The sulfur contents of PEX, SDD, IET and DDA are 40.25%, 37.42%, 21.77% and 22.85%, respectively. The high sulfur contents of PEX and SDD may well elucidate their large amount of generated $CS_2$ byproduct.

Additionally, the generation mechanisms of $CS_2$ from thiol collectors might also significantly influence its amount. However, up to now, the reports on the decomposition mechanisms of thiol collectors by the AOPs are scare. For the ozonation of xanthates, it is suggested that the attack on the C−O bond of xanthates by OH• radicals and nucleophilic reactions occurring for the C=S bond of the −CSS$^-$ group could result in the generation of the $CS_2$ [24,39]. In this case, the C−O bond in the PEX and C−N bond in the SDD have high electron density, which may be preferentially attacked by OH• radicals. Thus, the $CS_2$ may be generated from the PEX and SDD under $UV_{185+254\ nm}$ irradiation according to equations (3.4) and (3.5), respectively. As presented in table 1, the IET and DDA had molecular structures that were different to the PEX and SDD. The pathways of $CS_2$ generated from IET and DDA may be different from that for the PEX and SDD. In our future work, attention should be paid to elucidating the generation pathways of sulfur byproducts

$$CH_3CH_2OCSSK + OH\bullet + 1/2O_2 \rightarrow CH_3CH_2OH + CS_2 + K^+ + O_2^{\bullet-} \tag{3.4}$$

and

$$(CH_3CH_2)_2NCSSNa + OH\bullet + 1/2O_2 \rightarrow (CH_3CH_2)_2NH + CS_2 + K^+ + O_2^{\bullet-} \tag{3.5}$$

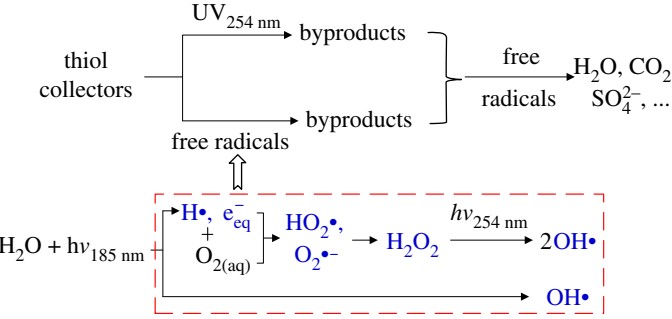

**Figure 7.** Proposed degradation mechanisms of thiol collectors under $UV_{185+254\ nm}$ irradiation.

## 3.5. The mechanisms of $UV_{185+254\ nm}$ photolysis of thiol collectors

As shown in figure 2 and table 2, thiol collectors could be degraded by both $UV_{254\ nm}$ and $UV_{185+254\ nm}$ photolysis, but with higher degradation rate constants under $UV_{185+254\ nm}$ irradiation. As mentioned above, free radicals such as OH• can be effectively generated in air purged water under 185 nm VUV irradiation as given in equations (1.1)−(1.6). OH• radicals, with an oxidation–reduction standard potential of 2.80 V, are non-selective and vigorous oxidants. The rate constants for OH• reacting with most organic compounds are within the range of $10^{6}-10^{9}\ 1/(mol\ s)$ [43]. Therefore, under $UV_{185+254\ nm}$ irradiation, two degradation mechanisms as shown in figure 7, i.e. direct $UV_{254\ nm}$ photolysis and indirect oxidation with free radicals such as OH•, should be responsible for the decomposition of collectors [44].

Nevertheless, the contributions of $UV_{254\ nm}$ photolysis and indirect oxidation with free radicals to collector decomposition were quite dependent on molecular structures of thiol collectors. For the PEX, SDD and DDA, both $UV_{254\ nm}$ photolysis and indirect oxidation with free radicals had contributed greatly as the $k_{collector}$ values under $UV_{185+254\ nm}$ irradiation were $1.60-2.63$ times higher than those under $UV_{254\ nm}$ irradiation. But for the IET, the $k_{IET}$ for $UV_{185+254\ nm}$ photolysis was 10.03 times higher than that for $UV_{254\ nm}$ photolysis. This suggested that indirect oxidation with free radicals was the main mechanism for the $UV_{185+254\ nm}$ photolysis of IET.

## 4. Conclusion

Thiol collectors (PEX, SDD, IET and DDA) could be effectively degraded by the $UV_{185+254\ nm}$ photolysis without dosing any oxidant. The removal efficiencies of PEX, SDD and IET reached nearly 100% upon 75 min of $UV_{185+254\ nm}$ irradiation. The $k_{collector}$ for four collectors decreased in the order $k_{SDD} > k_{PEX} > k_{IET} \gg k_{DDA}$. The DDA was the typical refractory flotation collector for $UV_{185+254\ nm}$ photolysis. In the $UV_{185+254\ nm}$ photolysis of the PEX and DDA, the $k_{collector}$ values were decreased at high initial pH, indicating neutral pH of flotation wastewaters can facilitate the collector decomposition. After 120 min of $UV_{185+254\ nm}$ irradiation, the $\gamma_c$ and $\gamma_s$ for four collectors reached $15-45\%$ and $25-75\%$, respectively, with the mineralization extent of PEX > SDD > IET > DDA. The effective degradation of thiol collectors was attained under $UV_{254\ nm}$ irradiation alone. Thus, two mechanisms, i.e. direct $UV_{254\ nm}$ photolysis and indirect oxidation with free radicals such as OH•, were responsible for the decomposition of collectors by the $UV_{185+254\ nm}$ photolysis.

After $UV_{185+254\ nm}$ irradiation for 120 min, the percentage of gaseous sulfur was 1.57% for PEX, 4.85% for SDD, 0.53% for IET and 0.48% for DDA, respectively, indicating that only a small fraction of sulfur in collectors was released into emission gas. For each collector, the amount of emitted $CS_2$ in gas phase was larger than that of $H_2S$. The aqueous $CS_2$ concentration increased rapidly in the first $10-20$ min for each collector, and was then reduced to a low level of $0.05-0.1\ mg\ l^{-1}$ at 120 min under $UV_{185+254\ nm}$ irradiation.

Data accessibility. The datasets supporting this article have been uploaded as part of the electronic supplementary material.

Authors' contributions. G.L. and X.W. set up the degradation system. G.L. and X.L. carried out the UV photolysis of thiol collectors. B.L. and X.W. measured the aqueous concentration of $CS_2$ and amount of emitted $CS_2$ and $H_2S$ in gas phase. P.F., G.L. and X.L. contributed to analysis data and wrote the draft of the manuscript. Finally, P.F. revised the manuscript.

Competing interests. We declare that there are no competing interests.

Funding. This work was financially supported by National Key R&D Program of China (no. 2018YFC1900604) and the National Natural Science Foundation of China (no. 51674017).

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
