## [Reviewer comments · Royal Society Open Science]

Review History

RSOS-190123.R0 (Original submission)

Review form: Reviewer 1 (Sayed Murtaza)

Is the manuscript scientifically sound in its present form?

Yes

Are the interpretations and conclusions justified by the results?

Yes

Is the language acceptable?

Yes

Is it clear how to access all supporting data?

Yes

Do you have any ethical concerns with this paper?

No

Have you any concerns about statistical analyses in this paper?

No

Recommendation?

Major revision is needed (please make suggestions in comments)

Comments to the Author(s)

The manuscript by Fu et al., deals with the The decomposition of toxic flotation reagents upon UV185+254nm irradiation was attractive due to operational simplicity and no dosage of oxidants. In this work, the degradation of typical thiol collectors (potassium ethyl xanthate (PEX), sodium diethyl dithiocarbamate (SDD), O-isopropyl-N-ethyl thionocarbamate (IET) and dianilino dithiophosphoric acid (DDA)) was investigated by the UV185+254nm photolysis. The degradation efficiencies and mineralization extents of collectors were assessed. The formation of CS₂ and H₂S byproducts was studied, and the mechanisms of collector degradation were proposed under UV185+254nm irradiation. The PEX, SDD and IET were decomposed with nearly 100% removal upon 75 min of UV185+254nm irradiation. The decomposition rate constants decreased in the order SDD>PEX>IET>>DDA, and the DDA was the refractory collector. After 120 min of UV185+254nm irradiation, 15–45% of carbon and 25–75% of sulfur of collectors were completely mineralized, and the mineralization extent decreased in the order PEX>SDD>IET>DDA. The percentage of gaseous sulfur (CS₂ and H₂S) was ranged from 0.48% to 4.85% for four collectors, showing the fraction of emitted sulfur byproducts was small. The aqueous CS₂ concentration increased in first 10–20 min, and was decreased to a low level of 0.05–0.1 mg/L at 120 min. Two mechanisms i.e., direct UV254nm photolysis and indirect oxidation with free radicals, were responsible for collector decomposition in the UV185+254nm photolysis. Overall, the manuscript is written in a good way, but i have cerertain suggestions/comments.

1. The authors are asked to update the literarature by some latest research some examples are;

(a) "Hydroxyl and sulfate radical mediated degradation of ciprofloxacin using nano zerovalent manganese catalyzed S₂O₈²⁻." *Chemical Engineering Journal* 356 (2019): 199-209.

(b) Narrowing the band gap of TiO₂ by co-doping with Mn²⁺ and Co²⁺ for efficient photocatalytic degradation of enoxacin and its additional peroxidase like activity: A mechanistic approach. *Journal of Molecular Liquids*, 272, pp.403-412.

(c) Solar light driven degradation of norfloxacin using as-synthesized Bi³⁺ and Fe²⁺ Co-doped ZnO with the addition of HSO₅⁻: Toxicities and degradation pathways investigation. *Chemical Engineering Journal*, 351, pp.841-855.

(d) Oxidative removal of brilliant green by UV/S₂O₈²⁻, UV/HSO₅⁻ and UV/H₂O₂ processes in aqueous media: A comparative study. *Journal of hazardous materials*, 357, pp.506-514.

2. did the authors have studied any toxicity of the sulfur degardation by products?.

3. How the authors are sure about the formation of OH (hydroxyl radicals) during the process. ?. Any ESR, EPR spectra??.

Review form: Reviewer 2

Is the manuscript scientifically sound in its present form?

Yes

Are the interpretations and conclusions justified by the results?

Yes

Is the language acceptable?

Yes

Is it clear how to access all supporting data?

Yes

Do you have any ethical concerns with this paper?

No

Have you any concerns about statistical analyses in this paper?

No

Recommendation?

Major revision is needed (please make suggestions in comments)

Comments to the Author(s)

Manuscript: RSOS-190123

Dear editor,

The manuscript evaluated the decomposition of toxic flotation reagents upon UV185+254nm irradiation and present some interesting results. Nevertheless, the results need to be better discussed. Some highlights that need improvement are:

Equations 8 and 11 need references.

How was the $t_{1/2}$ calculated?

The authors should standardize the nomenclature. Sometimes it is written UV185+254nm and sometimes VUV.

Why were the times of 25 and 75 min chosen to be highlighted in table 2?

The authors mention many times that the molecular structure of the collectors are different, so why was the effect of initial pH evaluated only for PEX and DDA?

Figure 4 should be changed to C/C_0 , so it would be easier to visualize the discussed results. The way it is presented, it is hard to compare them because of the scale of the four graphs.

The authors should take care of the scales in all graphs.

The experiments presented in Figure 4 were performed in which pH?

The authors could add equations regarding the possible reaction of mineralization for the four collectors. By adding this, it is easier to understand why using sulfate ions as an indicator of mineralization.

Experimental conditions should be added in figure's captions.

Review form: Reviewer 3

Is the manuscript scientifically sound in its present form?

Yes

Are the interpretations and conclusions justified by the results?

Yes

Is the language acceptable?

Yes

Is it clear how to access all supporting data?

Yes

Do you have any ethical concerns with this paper?

No

Have you any concerns about statistical analyses in this paper?

No

Recommendation?

Accept as is

Comments to the Author(s)

It is necessary to investigate new methods to remove chemicals from the floatation wastewater. Authors used UV254+185nm method to degrade a series of flotation chemicals, the mineralization, the by-products were well investigated, and the manuscript was well written. Thus, I think it can be accepted.

Review form: Reviewer 4

Is the manuscript scientifically sound in its present form?

Yes

Are the interpretations and conclusions justified by the results?

Yes

Is the language acceptable?

Yes

Is it clear how to access all supporting data?

Yes

Do you have any ethical concerns with this paper?

No

Have you any concerns about statistical analyses in this paper?

No

Recommendation?

Accept with minor revision (please list in comments)

Comments to the Author(s)

This paper in detail describes photolytic degradation of toxic flotation reagents. It deals with determining optimal conditions for the degradation, mechanism of degradation and monitoring certain degradation products. Gaseous byproducts were quantitatively determined which to the present knowledge there is no available work in the literature. This detailed approach to the solution of the environmental issue represents valuable information and therefore after minor revision can be published.

Specific comments

- There are some typing errors that should be corrected
- Line 90: not scare but scarce
- Line 91: replace "be different from each other" with differ
- Line 92: "sulfur is a key chemical element in functional groups..."
- Line 93 – 94: can you please explain the term organic sulfur because it appears later in the text
- Line 96: "... receives rare concern..." please rewrite to be more clear
- Line 140: "for" is unclear
- Line 165: is the chosen concentration of 100 mg/L near to the real concentration of collectors in the tailing pond? Why did you choose this concentration as the starting concentration?
- Line 150 – 151: in the name of the Table 1 remove "of thiol collectors"
- Line 228: excited instead exciting
- Line 247: based on the findings in this section did you from this point forward used only UV 245-185 nm in your experiments? If so, please add that comment to the end of this section.
- Line 257: in this section you choose onsl PEX and DDA for investigating the effect of initial pH, is there a reason for that?
- Line 460: did you state the pH of the investigated solution previously in the text? You stated that the wastewaters' pH is neutral to base, were these conditions for the experiment where you monitored k?

Decision letter (RSOS-190123.R0)

19-Mar-2019

Dear Professor Fu:

Title: UV_{185+254nm} photolysis of typical thiol collectors: Decomposition efficiency, mineralization and formation of sulfur byproducts

Manuscript ID: RSOS-190123

The editor assigned to your manuscript has now received comments from reviewers. We would like you to revise your paper in accordance with the referee and Subject Editor suggestions which can be found below (not including confidential reports to the Editor). Please note this decision does not guarantee eventual acceptance.

Please submit your revised paper before 11-Apr-2019. Please note that the revision deadline will expire at 00.00am on this date. If we do not hear from you within this time then it will be assumed that the paper has been withdrawn. In exceptional circumstances, extensions may be possible if agreed with the Editorial Office in advance. We do not allow multiple rounds of revision so we urge you to make every effort to fully address all of the comments at this stage. If deemed necessary by the Editors, your manuscript will be sent back to one or more of the original reviewers for assessment. If the original reviewers are not available we may invite new reviewers.

To revise your manuscript, log into <http://mc.manuscriptcentral.com/rsos> and enter your Author Centre, where you will find your manuscript title listed under "Manuscripts with Decisions." Under "Actions," click on "Create a Revision." Your manuscript number has been

appended to denote a revision. Revise your manuscript and upload a new version through your Author Centre.

Please also include the following statements alongside the other end statements. As we cannot publish your manuscript without these end statements included, if you feel that a given heading is not relevant to your paper, please nevertheless include the heading and explicitly state that it is not relevant to your work.

- Ethics statement

Please clarify whether you received ethical approval from a local ethics committee to carry out your study. If so please include details of this, including the name of the committee that gave consent in a Research Ethics section after your main text. Please also clarify whether you received informed consent for the participants to participate in the study and state this in your Research Ethics section.

OR

Please clarify whether you obtained the necessary licences and approvals from your institutional animal ethics committee before conducting your research. Please provide details of these licences and approvals in an Animal Ethics section after your main text.

OR

Please clarify whether you obtained the appropriate permissions and licences to conduct the fieldwork detailed in your study. Please provide details of these in your methods section.

- Acknowledgements

RSC Associate Editor:
Comments to the Author:

(There are no comments.)

RSC Subject Editor:

Comments to the Author:

(There are no comments.)

Reviewers' Comments to Author:

Reviewer: 1

Comments to the Author(s)

The manuscript by Fu et al., deals with the The decomposition of toxic flotation reagents upon UV185+254nm irradiation was attractive due to operational simplicity and no dosage of oxidants. In this work, the degradation of typical thiol collectors (potassium ethyl xanthate (PEX), sodium diethyl dithiocarbamate (SDD), O-isopropyl-N-ethyl thionocarbamate (IET) and dianilino dithiophosphoric acid (DDA)) was investigated by the UV185+254nm photolysis. The degradation efficiencies and mineralization extents of collectors were assessed. The formation of CS₂ and H₂S byproducts was studied, and the mechanisms of collector degradation were proposed under UV185+254nm irradiation. The PEX, SDD and IET were decomposed with nearly 100% removal upon 75 min of UV185+254nm irradiation. The decomposition rate constants decreased in the order SDD>PEX>IET>>DDA, and the DDA was the refractory collector. After 120 min of UV185+254nm irradiation, 15–45% of carbon and 25–75% of sulfur of collectors were completely mineralized, and the mineralization extent decreased in the order PEX>SDD>IET>DDA. The percentage of gaseous sulfur (CS₂ and H₂S) was ranged from 0.48% to 4.85% for four collectors, showing the fraction of emitted sulfur byproducts was small. The aqueous CS₂ concentration increased in first 10–20 min, and was decreased to a low level of 0.05–0.1 mg/L at 120 min. Two mechanisms i.e., direct UV254nm photolysis and indirect oxidation with free radicals, were responsible for collector decomposition in the UV185+254nm photolysis. Overall, the manuscript is written in a good way, but i have cerertain suggestions/comments.

1. The authors are asked to update the literarature by some latest research some examples are;

(a) "Hydroxyl and sulfate radical mediated degradation of ciprofloxacin using nano zerovalent manganese catalyzed S₂O₈²⁻." *Chemical Engineering Journal* 356 (2019): 199-209.

(b) Narrowing the band gap of TiO₂ by co-doping with Mn²⁺ and Co²⁺ for efficient photocatalytic degradation of enoxacin and its additional peroxidase like activity: A mechanistic approach. *Journal of Molecular Liquids*, 272, pp.403-412.

(c) Solar light driven degradation of norfloxacin using as-synthesized Bi³⁺ and Fe²⁺ Co-doped ZnO with the addition of HSO₅⁻: Toxicities and degradation pathways investigation. *Chemical Engineering Journal*, 351, pp.841-855.

(d) Oxidative removal of brilliant green by UV/S₂O₈²⁻, UV/HSO₅⁻ and UV/H₂O₂ processes in aqueous media: A comparative study. *Journal of hazardous materials*, 357, pp.506-514.

2. did the authors have studied any toxicity of the sulfur degardation by products?.

3. How the authors are sure about the formation of OH (hydroxyl radicals) during the process. ?. Any ESR, EPR spectra??.

Reviewer: 2

Comments to the Author(s)

Manuscript: RSOS-190123

Dear editor,

The manuscript evaluated the decomposition of toxic flotation reagents upon UV185+254nm irradiation and present some interesting results. Nevertheless, the results need to be better discussed. Some highlights that need improvement are:

Equations 8 and 11 need references.

How was the $t_{1/2}$ calculated?

The authors should standardize the nomenclature. Sometimes it is written UV185+254nm and sometimes VUV.

Why were the times of 25 and 75 min chosen to be highlighted in table 2?

The authors mention many times that the molecular structure of the collectors are different, so why was the effect of initial pH evaluated only for PEX and DDA?

Figure 4 should be changed to C/C_0 , so it would be easier to visualize the discussed results. The way it is presented, it is hard to compare them because of the scale of the four graphs.

The authors should take care of the scales in all graphs.

The experiments presented in Figure 4 were performed in which pH?

The authors could add equations regarding the possible reaction of mineralization for the four collectors. By adding this, it is easier to understand why using sulfate ions as an indicator of mineralization.

Experimental conditions should be added in figure's captions.

Reviewer: 3

Comments to the Author(s)

It is necessary to investigate new methods to remove chemicals from the floatation wastewater. Authors used UV254+185nm method to degrade a series of flotation chemicals, the mineralization, the by-products were well investigated, and the manuscript was well written. Thus, I think it can be accepted.

Reviewer: 4

Comments to the Author(s)

This paper in detail describes photolytic degradation of toxic flotation reagents. It deals with determining optimal conditions for the degradation, mechanism of degradation and monitoring certain degradation products. Gaseous byproducts were quantitatively determined which to the present knowledge there is no available work in the literature. This detailed approach to the solution of the environmental issue represents valuable information and therefore after minor revision can be published.

Specific comments

- There are some typing errors that should be corrected
- Line 90: not scare but scarce
- Line 91: replace "be different from each other" with differ
- Line 92: "sulfur is a key chemical element in functional groups..."
- Line 93 – 94: can you please explain the term organic sulfur because it appears later in the text
- Line 96: "... receives rare concern..." please rewrite to be more clear
- Line 140: "for" is unclear

- Line 165: is the chosen concentration of 100 mg/L near to the real concentration of collectors in the tailing pond? Why did you choose this concentration as the starting concentration?
- Line 150 – 151: in the name of the Table 1 remove “of thiol collectors”
- Line 228: excited instead exciting
- Line 247: based on the findings in this section did you from this point forward used only UV 245-185 nm in your experiments? If so, please add that comment to the end of this section.
- Line 257: in this section you choose onsl PEX and DDA for investigating the effect of initial pH, is there a reason for that?
- Line 460: did you state the pH of the investigated solution previously in the text? You stated that the wastewaters’ pH is neutral to base, were these conditions for the experiment where you monitored k?

Author's Response to Decision Letter for (RSOS-190123.R0)

See Appendix A.

Decision letter (RSOS-190123.R1)

15-Apr-2019

Dear Professor Fu:

Title: UV_{185+254nm} photolysis of typical thiol collectors: Decomposition efficiency, mineralization and formation of sulfur byproducts
Manuscript ID: RSOS-190123.R1

It is a pleasure to accept your manuscript in its current form for publication in Royal Society Open Science. The chemistry content of Royal Society Open Science is published in collaboration with the Royal Society of Chemistry.

Appendix A

Response to Referees

Reviewer: 1

Comments to the Author(s)

The manuscript by Fu et al., deals with the The decomposition of toxic flotation reagents upon UV185+254nm irradiation was attractive due to operational simplicity and no dosage of oxidants. In this work, the degradation of typical thiol collectors (potassium ethyl xanthate (PEX), sodium diethyl dithiocarbamate (SDD), O-isopropyl-N-ethyl thionocarbamate (IET) and dianilino dithiophosphoric acid (DDA)) was investigated by the UV185+254nm photolysis. The degradation efficiencies and mineralization extents of collectors were assessed. The formation of CS₂ and H₂S byproducts was studied, and the mechanisms of collector degradation were proposed under UV185+254nm irradiation. The PEX, SDD and IET were decomposed with nearly 100% removal upon 75 min of UV185+254nm irradiation. The decomposition rate constants decreased in the order SDD>PEX>IET>>DDA, and the DDA was the refractory collector. After 120 min of UV185+254nm irradiation, 15–45% of carbon and 25–75% of sulfur of collectors were completely mineralized, and the mineralization extent decreased in the order PEX>SDD>IET>DDA. The percentage of gaseous sulfur (CS₂ and H₂S) was ranged from 0.48% to 4.85% for four collectors, showing the fraction of emitted sulfur byproducts was small. The aqueous CS₂ concentration increased in first 10–20 min, and was decreased to a low level of 0.05–0.1 mg/L at 120 min. Two mechanisms i.e., direct UV254nm photolysis and indirect oxidation with free radicals, were responsible for collector decomposition in the UV185+254nm photolysis. Overall, the manuscript is written in a good way, but i have cerertain suggestions/comments.

Comment 1. The authors are asked to update the literarature by some latest research some examples are;

(a) "Hydroxyl and sulfate radical mediated degradation of ciprofloxacin using nano zerovalent manganese catalyzed S₂O₈²⁻." *Chemical Engineering Journal* 356 (2019): 199-209.

(b) Narrowing the band gap of TiO₂ by co-doping with Mn²⁺ and Co²⁺ for efficient photocatalytic degradation of enoxacin and its additional peroxidase like activity: A mechanistic approach. *Journal of Molecular Liquids*, 272, pp.403-412.

(c) Solar light driven degradation of norfloxacin using as-synthesized Bi³⁺ and Fe²⁺ Co-doped ZnO with the addition of HSO₅⁻: Toxicities and degradation pathways investigation. *Chemical Engineering Journal*, 351, pp.841-855.

(d) Oxidative removal of brilliant green by UV/S₂O₈²⁻, UV/HSO₅⁻ and UV/H₂O₂ processes in aqueous media: A comparative study. *Journal of hazardous materials*, 357, pp.506-514.

Response: The following four articles have been cited in the Introduction part.

[18] Shah NS., Khan JA, Sayed M, Khan ZU, Ali HS, Murtaza B, Khan HM, Imran M, Muhammad N. 2019 Hydroxyl and sulfate radical mediated degradation of ciprofl oxacin using nano zerovalent manganese catalyzed S₂O₈²⁻. *Chem. Eng. J.* **356**, 199–209. (doi: 10.1016/j.cej.2018.09.009)

[19] Rehman F, Sayed M, Khan JA, Shah NS, Khan HM, Dionysiou DD. 2018 Oxidative removal of brilliant green by UV/S₂O₈²⁻, UV/HSO₅⁻ and UV/H₂O₂ processes in aqueous media: A comparative study. *J. Hazard. Mater.* **357**, 506–514. (doi: 10.1016/j.jhazmat.2018.06.012)

[21] Sayed M, Arooj A, Shah NS, Khan JA, Shah LA, Rehman F, Arandiyan H, Khan AM, Khan AR. 2018 Narrowing the band gap of TiO₂ by co-doping with Mn²⁺ and Co²⁺ for efficient photocatalytic degradation of enoxacin and its additional peroxidase like activity: A mechanistic approach. *J. Mol. Liq.*, **272**, 403–412. (doi: 10.1016/j.molliq.2018.09.102)

[22] Shah NS, Khan JA, Sayed M, Khan ZU, Rizwan AD, Muhammad N, Boczkaj G, Murtaza B, Imran M, Khan HM, Zaman Gr. 2018 Solar light driven degradation of norfloxacin using as-synthesized Bi³⁺ and Fe²⁺ co-doped ZnO with the addition of HSO₅⁻: Toxicities and degradation pathways investigation. *Chem. Eng. J.* **351**, 841–855, (doi: 10.1016/j.cej.2018.06.111)

Comment 2. did the authors have studied any toxicity of the sulfur degradation by products?.

Response: At the current stage, we do not evaluate the toxicity of the sulfur degradation byproducts. Some toxic byproducts such as CS₂ may appear in the degradation of thiol collectors. So in the next work, we should investigate the acute and chronic toxicities of byproducts by using the ecological structure-activity relationship (ECOSAR) program.

Comment 3. How the authors are sure about the formation of OH (hydroxyl radicals) during the process. ?. Any ESR, EPR spectra??.

Response: In the previous work, H. Jablonowski et al. [26] have proved the formation of OH• radicals and superoxide anion (O₂⁻) in the VUV irradiation of water by the EPR spectroscopy. Thus, in the Line 106, the sentence “The formation of OH• radicals and superoxide anion (O₂⁻) in the VUV irradiation of water have been proved by the electron paramagnetic resonance (EPR) spectroscopy”.

[26] Jablonowski H, Bussiahn R, Hammer MU, Weltmann KD, von Woedtke T, Reuter S. 2015 Impact of plasma jet vacuum ultraviolet radiation on reactive oxygen species generation in bio-relevant liquids. *Phys. Plasmas* 22, 122008. (doi: 10.1063/1.4934989)

Reviewer: 2

Comments to the Author(s)

Manuscript: RSOS-190123

Dear editor,

The manuscript evaluated the decomposition of toxic flotation reagents upon UV185+254nm irradiation and present some interesting results. Nevertheless, the results need to be better discussed. Some highlights that need improvement are:

Comment 1. Equations 8 and 11 need references.

Response: The reference [31] has been cited for Equations 8. But the Equations 8 is defined by the authors for the first time to describe “the percentage of gaseous sulfur” in the degradation of thiol collectors.

[31] Fu PF, Lin XF, Li G, Chen ZH, Peng H. 2018 Degradation of thiol collectors using ozone at a low dosage: Kinetics, mineralization, ozone utilization, and changes of biodegradability and water quality parameters. Minerals, 8, 477. (doi:10.3390/min8110477)

Comment 2. How was the $t_{1/2}$ calculated?

Response: the pseudo-first-order kinetic equation is shown as Equation (1)

$$\ln \frac{C_t}{C_0} = -k \times t \quad (1)$$

where, C_0 and C_t (mg/L) are the collector concentration at initial and time t , k (min^{-1}) is the pseudo-first-order rate constant, and t (min) is degradation time. The half life ($t_{1/2}$) is defined as the degradation time that the C_t has been reduced to the value of $1/2C_0$. Thus, the half life ($t_{1/2}$) can be calculated by Equation (2)

$$t_{1/2} = -k \times \ln \frac{1}{2} \quad (2)$$

So, the half life ($t_{1/2}$) can be calculated from the pseudo-first-order rate constant k according to Equation (2).

Comment 3. The authors should standardize the nomenclature. Sometimes it is written UV185+254nm and sometimes VUV.

Response: In Table 2, the “VUV” is revised to “UV_{185+254nm}”.

In line 260, the “VUV” is revised to “UV_{185+254nm}”.

In some cases, the “185nm VUV” is used to replace “VUV” to differentiate VUV and UV_{185+254nm}.

Comment 4. Why were the times of 20 and 75 min chosen to be highlighted in table 2?

Response: As shown in Figure 2, the C_t/C_0 values for PEX, SDD and IET at 120 min are almost close to 100%. But at 20 and 75 min, the C_t/C_0 values for these collectors are quite different. So the degradation time of 20 and 75 min is chosen to compare the removal efficiency of four collectors with high distinguish degree.

Comment 5. The authors mention many times that the molecular structure of the collectors are different, so 6. why was the effect of initial pH evaluated only for PEX and DDA?

Response: As discussed in section 3.1, it can be seen that under UV_{185+254nm} irradiation, the degradation rate constants of four collectors have the order of SDD \approx PEX \gg IET $>$ DDA. So in the study of the effect of initial pH, we choose PEX as representative collector that can be effectively degraded, and select DDA as typical refractory collector among four collectors with different molecular structures.

Comment 6. Figure 4 should be changed to C/C0, so it would be easier to visualize the discussed results. The way it is presented, it is hard to compare them because of the scale of the four graphs.

Response: The scale of four graphs in Figure 4 has been changed as the following:

Figure 4 The variations of SO_4^{2-} concentration and relative concentrations of collector (PEX (a), SDD (b), IET (c) and DDA (d)), COD and TOC with irradiation time.

Comment 7. The authors should take care of the scales in all graphs.

Response: We have changed the scales of Figure 5. Now the Figure 5 is revised as:

Figure 5. The amounts of emitted CS_2 and H_2S in gas phase by the $UV_{185+254nm}$ photolysis of thiol collectors for 120 min.

Comment 8. The experiments presented in Figure 4 were performed in which pH?

Response: The experiments presented in Figure 4 were performed at initial pH of 10.0.

Comment 9. The authors could add equations regarding the possible reaction of mineralization for the four collectors. By adding this, it is easier to understand why using sulfate ions as an indicator of mineralization.

Response: At the current stage, the degradation pathways of sulfur element in four collectors are not clear. In addition, due to the different molecular structures of four collectors, the generation mechanism of SO_4^{2-} ions for four investigated collectors may be different from each other. So the addition of equations of SO_4^{2-} formation is not suitable in this article.

Comment 10. Experimental conditions should be added in figure's captions.

Response: Experimental conditions were added in the captions of Figure 2, Figure 3, Figure 4, Figure 5 and Figure 6. The captions of these six Figures are listed as following:

Figure 2. The variations of relative concentration (C_t/C_0) with irradiation time upon $\text{UV}_{254\text{nm}}$ (a) and $\text{UV}_{185+254\text{nm}}$ (b) irradiation. The insets are the pseudo-first-order kinetic fitting of $\ln(C_t/C_0)$ versus time t . Experimental conditions: collector concentration of 100 mg/L and initial pH of 10.0.

Figure 3. The variations of the k_{PEX} and k_{DDA} with initial pH under $\text{UV}_{185+254\text{nm}}$ irradiation. Experimental conditions: collector concentration of 100 mg/L.

Figure 4 The variations of SO_4^{2-} concentration and relative concentrations of collector (PEX (a), SDD (b), IET (c) and DDA (d)), COD and TOC with $\text{UV}_{185+254\text{nm}}$ irradiation time. Experimental conditions: collector concentration of 100 mg/L and initial pH of 10.0.

Figure 5. The amounts of emitted CS_2 and H_2S in gas phase by the $\text{UV}_{185+254\text{nm}}$ photolysis of thiol collectors for 120 min. Experimental conditions: collector concentration of 100 mg/L and initial pH of 10.0.

Figure 6. The variations of aqueous CS_2 concentration with $\text{UV}_{185+254\text{nm}}$ irradiation time. Experimental conditions: collector concentration of 100 mg/L and initial pH of 10.0.

Reviewer: 3

Comments to the Author(s)

It is necessary to investigate new methods to remove chemicals from the floatation wastewater. Authors used $\text{UV}_{254+185\text{nm}}$ method to degrade a series of flotation chemicals, the mineralization, the by-products were well investigated, and the manuscript was well written. Thus, I think it can be accepted.

Response: Thanks to the reviewer 3 for your comments.

Reviewer: 4

Comments to the Author(s)

This paper in detail describes photolytic degradation of toxic flotation reagents. It deals with determining optimal conditions for the degradation, mechanism of degradation and monitoring certain degradation products. Gaseous byproducts were quantitatively determined which to the present knowledge there is no available work in the literature. This detailed approach to the

solution of the environmental issue represents valuable information and therefore after minor revision can be published.

Specific comments

There are some typing errors that should be corrected

Comment 1. Line 90: not scare but scarce

Response: The word “scare” is changed to “scarce”.

Comment 2. Line 91: replace “be different from each other” with differ

Response: the “be different from each other” is replaced as “differ”.

Comment 3. Line 92: “sulfur is a key chemical element in functional groups...”

Response: “sulfur is a key chemical element in functional groups of thiol collectors” is revised as “the functional groups of thiol collectors have sulfur atoms”.

Comment 4. Line 93 – 94: can you please explain the term organic sulfur because it appears later in the text

Response: In this work, the organic sulfur refers to the S atoms in collector molecules. Due to the formation of CS₂, H₂S and SO₄²⁻ byproducts, the term “organic sulfur” of collectors is used to distinguish the inorganic sulfur byproducts such as H₂S and SO₄²⁻.

Comment 5. Line 96: “... receives rare concern...” please rewrite to be more clear

Response: “... receives rare concern...” is revised as “... receives little concern...”

Comment 6. Line 140: “for” is unclear

Response: the sentence “The irradiation intensity of UV_{254nm} for UV_{185+254nm} lamp was about 90% of that for UV_{254nm} lamp.” is revised as “The UV_{254nm} irradiation intensity of UV_{185+254nm} lamp was about 90% of UV_{254nm} lamp.”

Comment 7. Line 165: is the chosen concentration of 100 mg/L near to the real concentration of collectors in the tailing pond? Why did you choose this concentration as the starting concentration?

Response: The real collector concentration in tailing ponds is greatly affected with the dosage of collector in the mineral flotation process. When the tailings are discharged into the ponds, the real concentration of collectors may be as high as 50~150 mg/L. Therefore, we choose 100mg/L as the initial collector concentration in this work.

Comment 8. Line 150 – 151: in the name of the Table 1 remove “of thiol collectors”

Response: in the name of Table 1, the “of thiol collectors” is deleted.

Comment 9. Line 228: excited instead exciting

Response: the “exciting” is replaced with “excited”.

Comment 10. Line 247: based on the findings in this section did you from this point forward used only UV 245+185 nm in your experiments? If so, please add that comment to the end of this section.

Response: From section 3.2, only UV_{245+185 nm} is used in the experiments. In the section of “Conclusion”, the sentence “The DDA was the typical refractory flotation collector for UV_{185+254nm} photolysis.” has been added.

Comment 11. Line 257: in this section you choose only PEX and DDA for investigating the effect of initial pH, is there a reason for that?

Response: As discussed in section 3.1, it can be seen that under UV_{185+254nm} irradiation, the degradation rate constants of four collectors have the order of SDD>PEX>>IET>DDA. So in the study of the effect of initial pH, we choose PEX as representative collector that can be effectively degraded, and select DDA as typical refractory collector among four collectors with different molecular structures.

Comment 12. Line 460: did you state the pH of the investigated solution previously in the text? You stated that the wastewaters' pH is neutral to base, were these conditions for the experiment where you monitored k?

Response: In the section 3.2, we have investigated the effect of initial solution pH on the UV_{185+254nm} photolysis of PEX and DDA. The pH range for the experiments in section 3.2 was 7.0~12.0. Therefore, we have concluded that neutral pH of flotation wastewaters can facilitate the collector decomposition.